# Remaining Useful Life Prediction Across Conditions Based on a Health Indicator-Weighted Subdomain Alignment Network

**DOI:** 10.3390/s25154536

**Published:** 2025-07-22

**Authors:** Zhiqing Xu, Christopher W. K. Chow, Md. Mizanur Rahman, Raufdeen Rameezdeen, Yee Wei Law

**Affiliations:** Sustainable Infrastructure and Resource Management (SIRM), UniSA STEM, University of South Australia, Mawson Lakes, SA 5095, Australia; zhiqing.xu@mymail.unisa.edu.au (Z.X.); christopher.chow@unisa.edu.au (C.W.K.C.); mizanur.rahman@unisa.edu.au (M.M.R.); rameez.rameezdeen@unisa.edu.au (R.R.)

**Keywords:** bearing prognostics, health indicator, remaining useful life prediction, contrastive learning, subdomain adaptation

## Abstract

In recent years, domain adaptation (DA) has been extensively applied to predicting the remaining useful life (RUL) of bearings across conditions. Although traditional DA-based methods have achieved accurate predictions, most methods fail to extract multi-scale degradation information, focus only on global-scale DA, and ignore the importance of temporal weights. These limitations hinder further improvements in prediction accuracy. This paper proposes a novel model, called the health indicator-weighted subdomain alignment network (HIWSAN), which first learns feature representations at multiple scales, then constructs health indicators as temporal weights, and finally performs subdomain-level alignment. Two case studies based on the XJTU-SY and PRONOSTIA datasets were conducted, covering ablation, comparison, and generalization experiments to evaluate the proposed HIWSAN. Experimental results show that HIWSAN achieves an average MAE of 0.0989 and an average RMSE of 0.1189 across two datasets, representing reductions of 21.07% and 25.13%, respectively, compared to existing state-of-the-art methods.

## 1. Introduction

Bearing are critical to rotating machinery, as they are responsible for supporting shafts and reducing friction between moving parts [1]. Bearings faults can result in catastrophic equipment failures and serious safety risks [2]. Therefore, accurately predicting their remaining useful life (RUL) is crucial for industrial systems. RUL prediction methods can be classified into model-based methods and data-driven methods [3]. Model-based methods use physical or statistical modeling to capture failure mechanisms of bearings [4]. However, these methods are highly dependent on underlying assumptions, leading to significant prediction deviations if actual conditions differ from the assumptions. Additionally, the modeling process can be complex and time-consuming, requiring expert domain knowledge. In contrast, data-driven methods predict RUL by analyzing historical data, eliminating the need for expert knowledge.

Among data-driven methods, deep learning has been widely adopted due to its superior capability to handle nonlinear data. Widely used deep neural networks (DNNs), including convolutional neural network (CNN) [5], long short-term memory (LSTM) [6], gated recurrent unit (GRU) [7], temporal convolutional network (TCN) [8,9], and sample convolution and interaction network (SCINet) [10] have been applied to RUL prediction for effective feature extraction. In addition, combining signal processing techniques, such as frequency spectrum analysis [11], signal decomposition [12], and wavelet transforms [13], with DNNs has achieved success, although such methods require signal processing expertise.

Although these DNNs have achieved success, varying working conditions in industrial environments often lead to discrepancies in feature distributions between source and target domain data, making it difficult to achieve accurate RUL prediction using only DNNs [14,15].

To solve this problem, researchers have proposed domain adaptation (DA), which minimizes feature distribution discrepancies between two domains by learning transferable features. Traditional DA-based RUL prediction models involve feeding entire source and target domain data into a DNN for learning features, as shown in Figure 1a. These learned features are then mapped into a high-dimensional space, where domain-invariant features are extracted through discrepancy metrics, including maximum mean discrepancy (MMD) [8] and multi-kernel maximum mean discrepancy (MK-MMD) [15]. Although DA-based models achieve RUL prediction across different domains, they have three major limitations in improving prediction accuracy.

(1) Insufficient extraction of degradation information: Most existing methods feed the entire source domain and target domain data into a DNN, as shown in Figure 1a, causing the DNN to learn features at the same scale repeatedly. This process ignores the multi-scale information in the time series data, failing to thoroughly capture important details and features related to bearing degradation. For example, in a temperature prediction task, short-term temperature fluctuations may be related to weather changes and hourly effects, while long-term trends may be related to seasonal changes and annual effects [16]. Such multi-scale information is crucial for accurately representing degradation in industrial systems.

(2) Ignoring local-scale domain alignment: Features learned from both source and target domains are aligned in a high-dimensional space using a discrepancy metric, as shown in Figure 1a. This process aims to align discrepancies in feature distributions between source and target domains globally, as shown in Figure 2a. Since bearing degradation occurs in different stages, the feature distribution should account for multiple subdomains [17]. However, most methods only achieve the alignment of the feature distribution on a global scale, ignoring the discrepancies among subdomains at local scales.

(3) Lack of temporal weights: Not all time steps contribute equally during the process of DA. Time steps closer to failure typically provide more information related to bearing degradation. For example, a bearing performs well for the first 90% of the time, but fails in the last 10%. In such cases, the latter data should be assigned higher weights during DA, as it is more indicative of impending failure, while the earlier data are less critical.

To address these three limitations, this article proposes a novel RUL prediction model, called the health indicator-weighted subdomain alignment network (HIWSAN), which consists of three core components: an Encoder, a health indicator (HI) generator, and a Predictor. First, the Encoder captures fine-grained feature representations from raw data at multiple scales, as illustrated in Figure 1b. Next, the HI generator constructs HIs and uses these HIs to divide subdomains. Finally, HIs are treated as temporal weights and integrated into the Predictor to achieve subdomain alignment and RUL prediction.

The contributions of this research can be highlighted as follows:HIWSAN captures feature representations that reflect bearing degradation patterns. Applying these representations can achieve a series of prognostics tasks, including but not limited to HI construction and RUL prediction.HIWSAN achieves precise subdomain adaptation (SDA) by minimizing feature distribution discrepancies among subdomains, enhancing RUL prediction accuracy.HIWSAN leverages normalized HIs as temporal weights, enabling SDA to focus more attention on the alignment of degradation features.

The paper is organized as follows: Section 2 presents related work on multi-scale feature learning and DA for RUL prediction. Section 3 details the implementation methods of the proposed HIWSAN. Section 4 and Section 5 describe two case studies using HIWSAN to construct HI and predict RUL, while evaluating the model using standard metrics. Section 6 discusses the potential limitations of HIWSAN. Finally, Section 7 concludes.

## 2. Related Work

In the context of bearing RUL prediction, a significant amount of research has already been reported on multi-scale feature learning and DA. For example, querying Web of Science with “AB=(bearings) AND (AB=(RUL))” returned more than 200 results, highlighting the central role that multi-scale feature learning and DA play in RUL prediction models. Thus, this section focuses on these two areas for a brief literature review.

### 2.1. Multi-Scale Feature Learning for RUL Prediction

RUL prediction models aim to learn multi-scale features from input data and map them to predicted labels. Although LSTM [18], Transformer [19], and GRU [20] have demonstrated potential in feature learning for RUL prediction, CNNs and their variants have emerged as the most widely adopted architectures. This is due to their ability to be stacked into deep structures and their flexibility in using variable kernel sizes. The following parts review three popular CNN-based multi-scale feature learning networks.

**Convolutional auto-encoder (CAE)** is an unsupervised learning network that encodes high-dimensional input into low-dimensional feature representations. It captures both local and global features through multiple convolutional layers, enabling accurate RUL prediction. Researchers have used CAE to establish HIs, which are then used to predict RUL [21,22]. CAE can also be used for failure behavior classification. For example, Dong et al. [23] used CAE for failure behavior classification, then aligned the learned features within the same failure behavior, achieving accurate RUL prediction. However, during the feature extraction process, CAE may lose some critical degradation information, especially when dealing with complex degradation behaviors [24].

**TCN** captures bearing degradation features through the causal convolution and the dilated convolution [25]. The former prevents information leakage during feature learning, while the latter expands the receptive field to capture long-sequence information. Researchers have used TCN to capture bearing degradation features, followed by feature alignment between source and target domains, forming a representative approach for RUL prediction [26,27]. TCN can also be combined with other networks. For example, Wang et al. [28] combined features learned from a CNN and a TCN for RUL prediction.

**SCINet** downsamples raw data into two sub-sequences, which are then processed using distinct convolutional filters to extract features. Finally, interactive learning is used to compensate for the information loss during downsampling process [10]. This downsampling–convolution–compensation process can be encapsulated in a block and used recursively, allowing researchers to employ varying numbers of blocks to achieve RUL prediction [29,30]. Additionally, SCINet can be improved, as some scholars enhanced it with attention mechanisms or signal processing techniques for RUL prediction [31,32]. However, as the levels of SCINet tree structure increase, it faces challenges in effectively transferring information to deeper levels [33].

### 2.2. Domain Adaptation for RUL Prediction

In industrial settings, bearings exhibit varying degradation trends due to manufacturing differences and installation locations, leading to distinct feature distributions between bearings, as shown in Figure 1. Consequently, RUL prediction models trained on source domain data generally struggle to generalize to target domain data. For this problem, DA is applicable, as it aligns discrepancies in feature distributions between source and target domains, enhancing RUL prediction in unfamiliar settings. Traditional DA methods usually focus on global distribution alignment, ignoring the fine-grained information of subdomains. Recent works have focused on SDA, which aims for precise subdomain alignment. The following parts briefly review traditional DA methods, and then compare several SDA methods.

**Traditional domain adaptation** focuses on global-scale distribution alignment, and its implementation methods are divided into two categories [34]: (1) adversarial-based methods, which match feature distributions through modifying feature representations rather than performing geometric transformations. Several popular models, such as domain adversarial networks [35], dynamic adversarial adaptation networks [36], and selective adversarial adaptation networks [37] have been developed. However, these adversarial-based models often result in complex network structures and hyperparameter tuning. To solve these problems, (2) metric-based methods have been developed. These methods easily minimize statistical moments to reduce domain discrepancies, using metrics such as MMD [23], MK-MMD [26] and Wasserstein distance (WD) [20].

**Subdomain adaptation** aligns the same subdomains between the source and target domains at local scales, as shown in Figure 2b. Achieving SDA in RUL prediction is challenging due to the continuous nature of RUL labels, making it difficult to clearly define the number of subdomains. Two common methods to divide subdomains include (1) manually defined methods, which pre-divide the data into several subdomains. For example, Wu et al. [34] divided the data into two subdomains and then used local maximum mean discrepancy (LMMD) for subdomain alignment. Similarly, Ding et al. [17] evenly divided the continuous labels into 10 segments and then aligned each segment. However, relying on manually defined subdomain divisions requires expert knowledge. To adaptively divide subdomains without expert knowledge, (2) similarity or distance measurement methods have been developed. For example, Zhang et al. [38] used MMD to measure the similarity between two samples for dividing subdomains, while Cao et al. [20] used dynamic time warping to align the source and target domain data.

## 3. The Proposed Method

As illustrated in Figure 3, the proposed HIWSAN is a two-stage model. In stage 1, the Encoder is pre-trained to learn feature representations that contain multi-scale information from raw data. In stage 2, both source and target domain data are fed into the trained Encoder to learn feature representations. Subsequently, these representations are split into two branches: one branch is input to an HI generator for temporal weights calculation and subdomain division, while another branch is fed into the Predictor for SDA and RUL prediction. This section details the implementation of HIWSAN. All frequently used symbols in this section are listed in Table 1.

### 3.1. Stage 1: Pre-Training the Encoder

The process of pre-training the Encoder is achieved by an unsupervised learning network based on contrastive learning, which represents raw data into representations containing multi-scale information. In this research, the Encoder aims to learn a function $f_{\theta }$ that transforms full-life bearing data $x\in R^{T\times F}$ into a feature representation $r\in R^{T\times M}$ that effectively captures degradation patterns, where the symbols θ,T,F,M are as defined in Table 1. $f_{\theta }$ is trained by the proposed Encoder and the training process involves four steps.

**Step 1: Random sampling** is to sample data segments from raw data, ensuring that the Encoder receives data with varying lengths to capture multi-scale features. For any full-life bearing data *x*, a data segment $x_{b:f}$ is randomly cropped, and then moved backward by random time steps to $x_{a}$ and moved forward by random time steps to $x_{g}$, as illustrated in Figure 3. The data segment $x_{a:f}$ is regarded as sample 1 and the data segment $x_{b:g}$ is regarded as sample 2.

**Step 2: Multi-scale feature learning** is achieved by the proposed Encoder. It consists of a fully connected layer and seven stacked residual dilated convolution blocks. Each block contains two 1D convolution layers for feature extraction, two GeLU activation functions, and a residual structure to avoid gradient vanishing or explosion. Note that the dilation parameters of dilated convolution in the *l*-th block are $2^{l-1}$, but each block shares the same dilation parameters and kernel size. After a series of convolution and GeLU operations, the feature representation $r_{a:f}$ of sample 1 and the feature representation $r_{b:g}$ of sample 2 are generated.

**Step 3: Positive-negative pairs construction** aims to compare the similarity and dissimilarity between sample pairs. As illustrated in Figure 4, the overlapping parts of $r_{a:f}$ and $r_{b:g}$ are selected, and then marked as *r* and $r^{\prime }$, respectively. Feature representations at the same time step are treated as positive pairs, such as $r_{c}$ and $r_{c}^{\prime }$ at time step *c*, and $r_{d}$ and $r_{d}^{\prime }$ at time step *d*. Feature representations at different time steps are treated as negative pairs, such as $r_{c}$ and $r_{d}^{\prime }$, $r_{c}$ and $r_{d}$.

**Step 4: Contrastive loss** guides the Encoder’s parameter updates by encouraging high similarity in positive pairs and low similarity in negative pairs. The contrastive loss function of bearings at time step *t* can be formulated as follows:(1)$$L_{1}=-\log\frac{\exp\left(r_{t}\cdot r_{t}^{\prime }\right)}{\sum_{t^{\prime }\in \Omega }\left\{\exp\left(r_{t}\cdot r_{t^{\prime }}^{\prime }\right)+I_{t\neq t^{\prime }}\exp\left(r_{t}\cdot r_{t^{\prime }}\right)\right\}},$$
where Ω is the set of time steps within the temporal overlap of $r_{a:f}$ and $r_{b:g}$, and I is the indicator function.

To reduce the interference of outliers, the contrastive loss is designed to be a hierarchical structure, as shown in Figure 4. After calculating the contrastive loss of the first level, max-pooling is used along the time axis of the feature representation, and then the contrastive loss of the next level is calculated. The number of time steps in the feature representation is halved at each level until it is compressed to 1, and the average loss of all levels is taken to be the final loss. The process of pre-training the Encoder is summarized in Algorithm 1.
**Algorithm 1** Pre-training Encoder  1:**procedure** Pre-training(rawdata)  2:    **for** *x* in rawdata **do**  3:        // Random Sampling:  4:        $x_{b:f}\leftarrow$ randomly crop *x*;  5:        $x_{a:f},x_{b:g}\leftarrow$ sample two subsamples $x_{b:f}$;  6:        // Feature Learning:  7:        $r_{a:f},r_{b:g}\leftarrow$ Encoder $x_{a:f},x_{b:g}$;  8:        $r,r^{\prime }\leftarrow$ cropped overlap between $r_{a:f}$ and $r_{b:g}$;  9:        // Calculate contrastive loss:10:        $L_{1}\leftarrow$ HierLoss(*r*, $r^{\prime }$)11:        Update model parameters using $L_{1}$12:    **end for**13:**end procedure**14:**function** HierLoss(*r*, $r^{\prime }$)15:    $L_{1}\leftarrow L\left(r,r^{\prime }\right)$;16:    d←1;17:    **while** timelength(r)>1 **do**18:        // Maxpool1d operates along the time axis:19:        $r,r^{\prime }\leftarrow$ maxpool1d(*r*, $r^{\prime }$, kernel_size = 2);20:        $L_{1}\leftarrow L_{1}+L\left(r,r^{\prime }\right)$;21:        d←d+1;22:    **end while**23:    $L_{1}\leftarrow L_{1}/d$;24:    **return** $L_{1}$25:**end function**

### 3.2. Stage 2: RUL Prediction

The RUL prediction is implemented through an HI generator, a Predictor, and the trained Encoder. The HI generator takes the encoded feature representations $r^{S}$ and $r^{T}$ to construct ${\mathrm{HI}}^{S}$ and ${\mathrm{HI}}^{T}$, and to divide the source and target domains into subdomains. The Predictor treats ${\mathrm{HI}}^{S}$ and ${\mathrm{HI}}^{T}$ as temporal weightings and integrates them into the LMMD module to align the prediction results between the source and target subdomains. The overall training process consists of five steps:

**Step 1: Feature representations** are obtained from the pre-trained Encoder. Source domain data $x^{S}$ and target domain data $x^{T}$ are input into the pre-trained Encoder to generate source feature representation $r^{S}$ and target feature representation $r^{T}$.

**Step 2: Health indicator construction** aims to reflect the bearing degradation trend. HI is constructed by calculating the WD between the feature representations at each time step and the feature representations at the initial time step. The WD measures the distance between two probability distributions, and its calculation formula is as follows:(2)$$W\left(u,v\right)=\underset{\gamma ∼\Pi (u,v)}{\inf}E_{(p,q)∼\gamma }\left[∥p-q∥\right]$$
where Π(u,v) represents all joint distributions whose marginals are *u* and *v*. (p,q)∼γ means sampling a pair from a distribution γ∈Π(u,v).

After calculating the distance $d_{i}$ between each time step and the first time step, min–max normalization in the range [0,1] [15] is applied to convert the $d_{i}$ into HI. The HI at step *i* is expressed as follows:(3)$${\mathrm{HI}}_{i}=\frac{a+\left(d_{i}-d_{min}\right)\left(b-a\right)}{d_{max}-d_{min}},$$
where a=0, b=1.

**Step 3: Subdomain division** is performed via K-means clustering [39,40], which clusters the feature representations into *c* groups by minimizing internal variance. In this research, *c* is set to 2, dividing the source domain ${\mathrm{HI}}^{S}$ and the target domain ${\mathrm{HI}}^{T}$ into healthy subdomains (${\mathrm{HI}}_{h}^{S}$, ${\mathrm{HI}}_{h}^{T}$), and degradation subdomains (${\mathrm{HI}}_{d}^{S}$, ${\mathrm{HI}}_{d}^{T}$).

**Step 4: Subdomain adaptation** is achieved by the Predictor, which includes two fully connected layers and an LMMD module. First, fully connected layers map the source feature representation $r^{S}$ and target feature representation $r^{T}$ into RUL prediction results ${\tilde{y}}^{S}$ and ${\tilde{y}}^{T}$. Next, LMMD maps ${\tilde{y}}^{S}$ and ${\tilde{y}}^{T}$ into a high-dimensional feature space, where multiple Gaussian kernel functions quantify the differences between distributions. The LMMD between ${\tilde{y}}^{S}$ and ${\tilde{y}}^{T}$ is calculated as follows:(4)$$\mathrm{LMMD}\left({\tilde{y}}^{S},{\tilde{y}}^{T}\right)=\sum_{c\in \{h,d\}}{∥\frac{1}{N_{c}^{S}}\sum_{i=1}^{N_{c}^{S}}{\mathrm{HI}}_{c}^{S}\cdot \phi \left({\tilde{y}}_{i,c}^{S}\right)-\frac{1}{N_{c}^{T}}\sum_{j=1}^{N_{c}^{T}}{\mathrm{HI}}_{c}^{T}\cdot \phi \left({\tilde{y}}_{j,c}^{T}\right)∥}_{H}^{2},$$
where c∈{h,d} represents the healthy and degradation subdomains; ${\mathrm{HI}}_{c}^{S}$ and ${\mathrm{HI}}_{c}^{T}$ denote the temporal weights for source and target domains in category *c*; $N_{c}^{S}$ and $N_{c}^{T}$ denote the number of samples in the source and target domains for category *c*; H denotes the reproducing kernel Hilbert space; and ϕ(·) is defined as a Gaussian kernel function.

**Step 5: Model parameters optimization** involves two optimization objectives: (1) the RUL prediction loss $L_{R}$ between the ground truth and predicted RUL, (2) the domain discrepancy loss $L_{D}$ between source and target domain prediction values.

For the first optimization objective, the prediction error $L_{R}$ is defined using the mean square error (MSE), a common loss function for regression tasks. The MSE is formulated as follows:(5)$$L_{R}=\frac{1}{N^{S}}\sum_{i=1}^{N^{S}}{\left({\tilde{y}}_{i}^{S}-y_{i}^{S}\right)}^{2},$$
where ${\tilde{y}}_{i}^{S}$ and $y_{i}^{S}$ denote RUL prediction values and the ground truth of the source domain. $N^{S}$ is the number of samples in the source domain.

For the second optimization objective, the domain discrepancy loss $L_{D}$ is defined using the LMMD in Equation (Equation 4). The total loss of the proposed HIWSAN is defined as follows:(6)$$L_{2}=L_{R}+\lambda L_{D}=\frac{1}{N^{S}}\sum_{i=1}^{N^{S}}{\left({\tilde{y}}_{i}^{S}-y_{i}^{S}\right)}^{2}+\lambda \sum_{c\in \{h,d\}}{\mathrm{HI}}_{c}\cdot \mathrm{LMMD}\left({\left\{{\tilde{y}}_{i}^{S}\right\}}_{i=1}^{N_{c}^{S}},{\left\{{\tilde{y}}_{j}^{T}\right\}}_{j=1}^{N_{c}^{T}}\right),$$
where λ denotes the tradeoff parameter; ${\mathrm{HI}}_{c}$ is the temporal weights associated with category *c*, and *c* is set to 2 in this research.

Once the loss function $L_{2}$ is defined, the optimal parameters of the HIWSAN can be searched. The calculation steps are summarized in Algorithm 2.
**Algorithm 2** RUL Prediction  1:**procedure** Main(raw data)  2:    **for** $x^{S},x^{T},y^{S}$ in rawdata **do**  3:        // Feature learning:  4:        $r^{S},r^{T}\leftarrow$ trained Encoder $x^{S},x^{T}$;  5:        // Subdomain division:  6:        ${\mathrm{HI}}^{S},{\mathrm{HI}}^{T}\leftarrow$ HI generator $r^{S},r^{T}$;  7:        ${\mathrm{HI}}_{h}^{S},{\mathrm{HI}}_{d}^{S},{\mathrm{HI}}_{h}^{T},{\mathrm{HI}}_{d}^{T}\leftarrow$ K-means ${\mathrm{HI}}^{S},{\mathrm{HI}}^{T}$;  8:        // RUL prediction:  9:        ${\left\{{\tilde{y}}_{i}^{S}\right\}}_{i=1}^{N^{S}},{\left\{{\tilde{y}}_{j}^{T}\right\}}_{j=1}^{N^{T}}\leftarrow$ Predictor $r^{S},r^{T}$;10:        // Calculate loss:11:        $L_{R}\leftarrow$ MSELoss$\left({\left\{{\tilde{y}}_{i}^{S}\right\}}_{i=1}^{N^{S}},{\left\{y_{i}^{S}\right\}}_{i=1}^{N^{S}}\right)$;12:        $L_{D}\leftarrow$ LMMDLoss$\left({\left\{{\tilde{y}}_{i}^{S}\right\}}_{i=1}^{N^{S}},{\left\{{\tilde{y}}_{j}^{T}\right\}}_{j=1}^{N^{T}},{\mathrm{HI}}_{h}^{S},{\mathrm{HI}}_{d}^{S},{\mathrm{HI}}_{h}^{T},{\mathrm{HI}}_{d}^{T}\right)$;13:        $L_{2}\leftarrow$$L_{R}$ + $\lambda L_{D}$;14:        Update model parameters using $L_{2}$15:    **end for**16:**end procedure**

## 4. Case Study 1: HI Construction

The aim of Case Study 1 is to validate the effectiveness of HIs constructed from the learned feature representations, ensuring they are monotonic, robust, and strongly correlated with actual bearing degradation.

Two open-source bearing datasets, namely XJTU-SY [41] and PRONOSTIA (IEEE PHM 2012) [42], are used in this case study and the next (see Section 5), because the bearing prognostics literature has standardized on these datasets. Using these public datasets facilitates reproducibility and fair comparisons of different prognostic methods.

Case Study 1 conducts ablation experiments and comparison experiments on the XJTU-SY dataset to compare different HI construction methods. Subsequently, the generalization performance of the proposed HI construction method is validated based on the PRONOSTIA dataset.

### 4.1. Data Description

The XJTU-SY bearing dataset comprises run-to-failure data of 15 bearings under three different conditions, as detailed in Table 2. The PRONOSTIA bearing dataset contains run-to-failure data of 17 bearings under three different conditions, as detailed in Table 3. Their sampling schemes are described in Figure 5.

### 4.2. Model Design

The Adam optimizer is selected, and the learning rate adjustment strategy adopts StepLR. The initial learning rate is 0.001, with a decay factor of 0.1. During training, the learning rate is adjusted every 50 epochs, and the maximum number of epochs is 200. The batch size is set to 1, meaning that the data for all time steps are input at once. The hyperparameter settings used for pre-training are summarized in Table 4. The fully connected layer first reduces the number of features of raw data to 64. After passing through seven dilated convolution blocks, the number of features increases to 320. Taking Bearing 1_2 in the XJTU-SY dataset as an example, Table 5 shows the architectural parameters of the proposed Encoder.

The proposed Encoder contains about 2.6 million trainable parameters, with an estimated total size of 33.95 MB, which is lightweight given the high dimensionality of the input data. By leveraging dilated convolutions, the Encoder is able to efficiently capture multi-scale temporal features with fewer operations than traditional convolutional networks. When running on an Intel Core i5-12500H processor, the model’s average inference time per sample is 11.32 milliseconds, achieving real-time or near-real-time processing without GPU acceleration. The combination of compact model size and fast inference time shows the potential of our approach for practical applications in industrial settings.

### 4.3. Evaluation Metrics for HI Construction

Monotonicity, correlation, robustness, and a comprehensive metric are used to quantitatively evaluate the performance of HIs. Polynomial fitting is first applied to decompose the HI into an average trend and a random part:(7)$$H\left(t\right)=H_{T}\left(t\right)+H_{R}\left(t\right),$$
where H(t) is the HI at time *t*, with $H_{T}\left(t\right)$ and $H_{R}\left(t\right)$ representing the average trend and the random part.

**Monotonicity** measures the mean absolute difference between the number of positive differentials and the number of negative differentials [43]:(8)$$\mathrm{Mon}=\frac{1}{N-1}\left|\mathrm{No}.\;\mathrm{of}\;\left(\Delta H_{T}\left(t\right)>0\right)-\mathrm{No}.\;\mathrm{of}\;\left(\Delta H_{T}\left(t\right)<0\right)\right|,$$
where *N* is the total number of $H_{T}\left(t\right)$ values, and $\Delta H_{T}\left(t_{n}\right)=H_{T}\left(t_{n+1}\right)-H_{T}\left(t_{n}\right)$. As an asset without maintenance can only degrade monotonically over time, a higher monotonicity value (0≤Mon≤1) is associated with better health indication.

**Correlation**, also known as *trendability* [44], evaluates the degree of correlation between the HI and bearing degradation status:(9)$$\mathrm{Corr}=\frac{\left|\sum_{n=1}^{N}\left(H\left(t_{n}\right)-\bar{H}\right)\left(t_{n}-\bar{T}\right)\right|}{\sqrt{\sum_{n=1}^{N}{\left(H\left(t_{n}\right)-\bar{H}\right)}^{2}\sum_{n=1}^{N}{\left(t_{n}-\bar{T}\right)}^{2}}},$$
where $\bar{H}=\frac{1}{N}\sum_{t=1}^{N}H\left(t_{n}\right)$, and $\bar{T}=\frac{1}{N}\sum_{n=1}^{N}t_{n}$. A higher correlation score indicates a strong correlation with the state of bearing degradation.

**Robustness** measures the ability of HI to resist random fluctuations [45]:(10)$$\mathrm{Rob}=\frac{1}{N}\sum_{n=1}^{N}\exp\left(-\left|\frac{H_{R}\left(t_{n}\right)}{H(t_{n})}\right|\right).$$

**Comprehensive metric**, also called hybrid metric, is a linear combination of the preceding metrics for assessing the overall ability of an HI [21]:(11)CM=0.4·Mon+0.3·Corr+0.3·Rob. The choice of weights above follows Chen et al.’s [21], but we acknowledge that different choices have been reported in the literature; for example, see [45]. There is currently no consensus on how Mon, Corr, and Rob should be weighted to form the comprehensive metric, but as the results in Section 4.5 show, the Mon metric is the most challenging and should be given the highest weight, just as Chen et al. [21] and Zhang et al. [45].

### 4.4. Ablation Experiments of the Proposed Encoder

To assess the proposed HI construction method, its performance is tested under condition 1 of the XJTU-SY dataset. Bearing 1_1 is used for training; Bearing 1_2, Bearing 1_3, and Bearing 1_5 are used for testing; and Bearing 1_4 is abandoned due to sudden failure. The proposed HI construction method consists of three key modules: (1) random sampling (RS), (2) residual dilated convolution (RDC), and (3) hierarchical contrastive (HC), so the structures of models for the ablation experiment are as follows:w/o RS: Model A omits the RS module and inputs all time-step data into the Encoder in each epoch.w/o RDC: Model B replaces 7 RDC blocks with 14 stacked normal 1-D convolution layers, keeping the kernel size and number of layers unchanged.w/o HC: Model C omits the HC loss, and only calculates one-level contrastive loss.Model D is the proposed Encoder.

All experiments are repeated 10 times to minimize randomness. The ablation experiment results for the four models in terms of Mon, Corr, Rob and CM can be seen in Table 6 and Figure 6.

According to Figure 6, the CM value of Model D is higher than that of the other models, indicating that HIs constructed from the proposed method most effectively reveal the bearing degradation trends. The CM value of Model A ranks last because omitting the RS module prevents the network from learning multi-scale features. Since bearing degradation includes both long-term trends and short-term patterns, omitting the RS module means the model learns at a single scale repeatedly, resulting in the loss of degradation information. The CM value of Model C surpasses only that of Model A, because the HC module assists the network in mitigating the impact of outliers by averaging the similarities between positive-negative pairs. Without the HC module, the contrastive loss is only calculated once, even with a few time steps sampled. Model B ranks second in terms of CM, suggesting that replacing normal 1-D convolution with the RDC module effectively improves the performance of the HIs.

### 4.5. Comparison with Related HI Construction Methods

HI construction typically involves two key steps: (1) dimensionality reduction; (2) similarity measurement. As similarity measurement generally involves simple distance computations between the current and initial states, the comparison focuses on different dimensionality reduction strategies. Two classic methods (principal component analysis (PCA)-HI, isometric mapping (ISOMAP)-HI), and three Encoder-based methods (auto-encoder (AE)-HI, MCAN-HI [22], and MSMHA-HI [46]) are chosen.

Condition 2 of the XJTU-SY dataset is used in this set of comparison experiments, where Bearing 2_1 is used to train the network; Bearing 2_2, Bearing 2_3, and Bearing 2_5 are used for testing; and Bearing 2_4 is discarded due to too few time steps. All experiments are repeated 10 times to minimize randomness. Table 7 shows the values of HI metrics associated with test bearings.

According to Table 7, the proposed-HI achieves the highest CM values on three bearings, indicating that feature representations learned from the Encoder are more effective for constructing HIs than raw data. This outcome aligns with the fact that learned representations capture more information than raw data. The CM values of MSMHA-HI and MCAN-HI rank second and third on three bearings, respectively. Both methods extract multi-scale coded features from raw data, further highlighting the importance of multi-scale features in HI construction. The CM values of AE-HI on three bearings rank fourth because an AE loses multi-scale degradation information during feature extraction. The CM values of PCA-HI and ISOMAP-HI rank last on three bearings, confirming the advantage of network-based methods in HI construction. In addition, the CM values for ISOMAP-HI are higher than those for PCA-HI, suggesting that the manifold learning method is more suitable for constructing HIs than the linear dimensionality reduction algorithm, since the degradation trend of bearings is not linear.

### 4.6. Validation of Model Generalization Performance

To assess the generalization ability of the proposed-HI, the PRONOSTIA dataset is used to construct HIs and divide their subdomains. For validation, the model parameters remain unchanged. Figure 7 shows time-domain curves, HIs and their subdomains for Bearing 1_3, Bearing 2_2, and Bearing 3_3. The proposed-HI is also compared with MCAN-HI [22] and MSMHA-HI [46]. As shown in Table 8, the average results of the proposed-HI outperform the others, demonstrating its superior generalization capability.

## 5. Case Study 2: RUL Prediction

In order to demonstrate that the proposed HIWSAN can achieve high RUL prediction accuracy, Case Study 2 conducts ablation and comparison experiments to compare the RUL prediction accuracy on the PRONOSTIA dataset, and introduces three evaluation metrics to quantify the accuracy. Subsequently, the generalization performance of the HIWSAN is assessed based on the XJTU-SY dataset.

### 5.1. Model Design

The parameter configuration during model training is the same as that in Table 4. In addition, the number of subdmains *c*, which is a user-defined parameter based on the domain distribution of data, is configured as 2, following Wu et al. [34]. The tradeoff parameter λ is set to 0.01. Taking Bearing 1_3 of the PRONOSTIA dataset as an example, it has a total of 2375 time steps, and each time step has 2560 features. First, raw data are input into the trained Encoder for learning feature representations, where the number of features is reduced from 2560 to 320. Next, the learned representations are divided into two branches, one is input into the HI generator for HI construction and subdomain division; the other is input into the Predictor for RUL prediction. Table 9 presents the transformations of the data shape throughout the proposed RUL prediction model, as outlined in Algorithm 2.

### 5.2. Evaluation Metrics for RUL Prediction

The accuracy of RUL prediction is measured using three types of prediction errors, namely mean absolute error (MAE), root mean square error (RMSE), and Score [42]:(12)$$\begin{array}{cc} \mathrm{MAE} & =\frac{1}{N}\sum_{i=1}^{N}\left|{\tilde{y}}_{i}-y_{i}\right|, \end{array}$$(13)$$\begin{array}{cc} \mathrm{RMSE} & =\sqrt{\frac{1}{N}\sum_{i=1}^{N}{\left({\tilde{y}}_{i}-y_{i}\right)}^{2}}, \end{array}$$(14)$$\begin{array}{cc} \mathrm{Score} & =\frac{1}{N-1}\sum_{i=1}^{N-1}A_{i}, \end{array}$$
where(15)$$\begin{array}{cc} A_{i} & =\left\{\begin{array}{cc} e^{-\ln\left(0.5\right)\cdot \left(Er_{i}/5\right)} & \mathrm{if}\;Er_{i}\leq 0,\\ e^{+\ln\left(0.5\right)\cdot \left(Er_{i}/20\right)} & \mathrm{if}\;Er_{i}>0, \end{array}\right. \end{array}$$(16)$$\begin{array}{cc} Er_{i} & =\frac{y_{i}-{\tilde{y}}_{i}}{y_{i}}\times 100\%, \end{array}$$
where $y_{i}$ and ${\tilde{y}}_{i}$ are the actual RUL and the predicted RUL, respectively. The rationale for using the thresholds 5 and 20 in (14) lies in penalizing late prediction (i.e., predicting a longer RUL than actual, resulting in a negative $Er_{i}$) more than early prediction (i.e., predicting a shorter RUL than actual, resulting in a positive $Er_{i}$), since early predictions cause premature maintenance, whereas late predictions can lead to major disruptions. For example, an early prediction with $Er_{i}=20\%$ obtains a score of 0.5, whereas a late prediction with $Er_{i}=-20\%$ obtains a much lower score of 0.125. The thresholds of 5 and 20 in (14) are subjective, but have been the de facto standard since the publication of the PRONOSTIA dataset [42].

### 5.3. Sensitivity Analysis of the Tradeoff Parameter

Given the sensitivity of the proposed HIWSAN to the trade-off parameter λ, its performance is evaluated over the range [0.001, 0.005, 0.01, 0.05, 0.1]. To reduce the effect of randomness, all experiments are repeated 10 times, and the results are presented in Figure 8.

The detailed results in Figure 8 show the trends for different tradeoff parameters λ in terms of RMSE and MAE. When λ increases from 0.001 to 0.01, the prediction accuracy improves. Smaller λ cannot fully emphasize $L_{D}$ in Equation (Equation 6), resulting in a significant difference between the feature distributions of the source and target domains, which harms the prediction. Increasing λ from 0.001 to 0.01 enhances $L_{D}$, making the distributions of the source and target domains closer, thereby improving the prediction accuracy. However, further increasing λ to 0.1 significantly reduces the accuracy. This is because $L_{D}$ dominates the optimization process, causing the model to focus too much on distribution alignment and ignore the main RUL prediction task, resulting in a decrease in prediction accuracy.

### 5.4. Ablation Experiments of Proposed HIWSAN

To verify that the proposed HIWSAN can achieve high prediction accuracy, six transfer tasks are completed based on the PRONOSTIA dataset, as detailed in Table 10. The HIWSAN consists of three core modules: (1) the SDA module, (2) the HI generator, and (3) the RDC-based Encoder. The structures of the models used in the ablation experiment are as follows:w/o SDA: Model A omits the SDA module, serving as a baseline model that feeds feature representations directly into the Predictor without domain alignment.w/o HI: Model B omits the HI generator, meaning that feature representations are fed into the Predictor to perform DA without temporal weights and SDA.w/o RDC: Model C replaces the RDC blocks of Encoder with normal convolution layers, meaning that the encoded feature representations do not contain multi-scale information.Model D is the proposed HIWSAN.

All experiments are repeated 10 times to minimize randomness. Table 11 presents the MAE, RMSE, and Score for the four models, and Figure 9 displays the average results of these models. RUL prediction results for six transfer tasks are presented in Figure 10.

According to Figure 9, the prediction errors of Model A are significantly higher than those of other models. Additionally, the prediction curves of Model A, as shown in Figure 10, are nearly horizontal, indicating that the Predictor without domain alignment is ineffective for RUL prediction. This is because the source and target domains have different feature distributions, making a model trained only on source data ineffective for accurate prediction on target data. The prediction errors of Model B are higher than those of Model C and Model D, because HIs as temporal weights help the model focus more attention on later time steps during DA. In addition, the K-means clustering accurately divides the subdomains. Without these two modules, Model B can not achieve fine-grained SDA. The prediction errors of Model C are only higher than those of Model D, meaning that extracting multi-scale degradation information can improve the prediction accuracy. The proposed Model D has the lowest prediction errors, and its prediction curves are closest to the ground truth, showing its effectiveness.

### 5.5. Comparison with Related RUL Prediction Methods

To evaluate the performance of the proposed HIWSAN, comparisons are made with two baseline models: TCNN proposed by Cheng et al. [8], which is based on MK-MMD; and WD-WDANN proposed by Hu et al. [47], which is based on DANN. Additionally, the HIWSAN is compared with two advanced DA methods: MADA proposed by Zhuang et al. [48] and MCDA proposed by Dong et al. [49], and two advanced SDA methods: DSAN-WM proposed by Wu et al. [34] and GSAN proposed by Zhuang et al. [50]. These models are tested across six transfer scenarios outlined in Table 10. The experimental results comparing the proposed RUL prediction model with related methods in terms of MAE, RMSE, and Score are presented in Table 12.

According to Table 12, the average prediction errors of SDA models are lower than those of DA models, indicating that subdomain alignment can improve prediction accuracy. Among SDA models, DSAN-WM has the highest errors, indicating that pseudo-labels introduce additional misclassification errors.

Among DA models, TCNN and WD-WDANN have the highest prediction errors. This is because baseline models only consist of a domain alignment module and a backbone network without introducing additional constraints. The prediction errors of WD-WDANN are lower than those of TCNN, indicating that the adversarial transfer network can better capture the mapping relationship between domains within the same transfer scenario. Both MADA and the proposed HIWSAN employ a contrastive learning framework. However, MADA incorporates the positive pairs matching module as a constraint within the loss function, whereas the HIWSAN uses the positive-negative pairs module for data preprocessing. This comparison suggests that data preprocessing combined with a simpler model is more reliable than employing a complex model structure. MCDA achieves conditional and marginal distribution alignment by introducing multiple constraints in the adversarial network. The HIWSAN outperforms MCDA, indicating that preprocessing data through the Encoder is more effective than adding multiple constraints.

### 5.6. Validation of Model Generalization Performance

To evaluate the generalization performance of the proposed HIWSAN, the XJTU-SY dataset is also used, and Table 13 defines the transfer tasks. The HIWSAN was compared with two other SDA models (DSAN-WM and GSAN).

The prediction curves shown in Figure 11 of the proposed HIWSAN are closer to the ground truth, and the prediction errors in Table 14 are the lowest, indicating the generalization performance of the proposed HIWSAN.

## 6. Discussion

This section provides details left out in the preceding sections and addresses potential concerns that may arise.

**Degradation scaling**: To address the limitation of traditional linear normalization in representing nonlinear degradation processes, a sigmoid scaling method and an exponential weighting method were introduced.

The sigmoid scaling method leverages the S-shaped curve of the sigmoid function to nonlinearly map the HI, thereby better capturing the non-uniform degradation trends in bearings. The sigmoid-based HI is computed as follows:(17)$${\mathrm{HI}}_{i}=\frac{1}{1+e^{-k\left(d_{i}-\mu \right)}},$$
where $d_{i}$ denotes the Wasserstein distance at time step *i*, μ is a parameter controlling the position of the inflection point (which is set to the mean of all distance values in this research), and *k* is a parameter controlling the steepness of the sigmoid curve. Larger *k* values produce a steeper curve, accentuating nonlinear scaling.The exponential weighting method highlights the accelerated degradation in the later stages of bearing health. The exponential-based HI is calculated as follows:(18)$${\mathrm{HI}}_{i}=\frac{1-e^{-\alpha \cdot d_{i}}}{1-e^{-\alpha \cdot d_{\max}}},$$
where $d_{i}$ is the Wasserstein distance at time step *i*, $d_{\max}$ is the maximum distance value, and α>0 controls the strength of exponential weighting. As α increases, normalization accentuates late-stage degradation more.

Figure 12 shows the HIs constructed by the three normalization methods. Although the sigmoid scaling and exponential weighting methods provide semantically reasonable representations of nonlinear degradation, both involve hyperparameter selection (*k*, μ, and α), which can significantly influence the shape and sensitivity of the resulting HI curves. For instance, the sigmoid method performs well when appropriate parameters are selected, as shown in Figure 12b, but poor parameter selection in the exponential weighting method can lead to catastrophic results, as shown in Figure 12c. To avoid the subjectivity and potential instability associated with manual tuning, the min–max normalization method offers a more stable and robust alternative, as it does not rely on any hyperparameter.

**Strong noise**: This is a concern as it can mask useful signal features, hampering extraction of accurate and reliable features from raw data. To assess the impact of strong noise, a set of experiments on the HI curves and HI metrics (see Section 4.3) were conducted, where zero-mean Gaussian noise was added to the raw data for Bearing 1_3 in the XJTU-SY dataset. The standard deviation of the noise was varied from 0.5 to 2 to represent four noise levels. The HI constructed using raw data were then compared to HIs constructed using the noisy data in terms of the associated HI curves (see Figure 13) and HI metrics (see Table 15).

Figure 13 shows that as the noise level increases, the HI curves become less monotonic, i.e., less indicative of bearing health. Table 15 shows that as the noise level increases, the values of the HI metrics decrease. It is not surprising that the proposed HI generator shows deteriorated performance in the presence of strong noise, which is consistent with most deep learning models [51]. This is why denoising is a vital step in data preprocessing. Fortunately, well-established signal processing techniques such as the Kalman filter can be used to filter out additive Gaussian noise.

**Model interpretability**: This refers to the intrinsic properties of a deep model measuring the degree to which the inference result of the deep model is predictable or understandable to human beings [52]. The understandability of a model depends on the human, and as such, model interpretability is often visualized in a human-friendly manner through an interpretation algorithm, rather than summarized as a number. For example, the popular interpretation algorithm Grad-CAM [53] provides visual explanations of a CNN, in the form of a map highlighting important regions of an image associated with the predicted class, based on gradient information flowing into the final convolutional layer of the CNN. Visualizations like those provided by Grad-CAM are relevant for computer vision applications, more so after the discovery of the vulnerability of deep neural networks to adversarial attacks [51].

For prognostics, the risk of adversarial attacks is small as test data are gathered in a controlled environment. It is less important for human users to visualize how a model summarizes condition-monitoring data into a human-friendly number representing the health status of the object, than whether this number is representative. Furthermore, while an interpretable model may more readily earn a user’s trust than a model that is not, the former does not necessarily score higher on the performance metrics (e.g., accuracy) than the latter [54]. Nevertheless, the construction of an HI can be thought of as a pursuit of model interpretability as it summarizes raw data into a human-interpretable HI, while the HI metrics (see Section 4.3) can be thought of as interpretation algorithms evaluating the trustworthiness of the HI.

**Model deployability**: In many industrial applications, such as condition monitoring or anomaly detection in rotating machinery, predictions must be made in real time or near-real time. Our model contains about 2.63 million trainable parameters, taking up about 33.95 MB of storage and supporting fast inference. Preliminary tests show that the model achieves an average inference latency of only 11.32 milliseconds per sample on an Intel Core i5-12500H processor, which is far lower than the real-time processing requirement of 50 milliseconds and meets the deployment criteria for many industrial applications.

## 7. Conclusions

This paper proposes a novel remaining useful life (RUL) prediction model called health indicator-weighted subdomain alignment network (HIWSAN), which comprises an Encoder, a health indicator (HI) generator, and a Predictor. The results of our ablation experiments, comparative experiments, and validation experiments, presented through two case studies, provide concrete evidence that (1) HIWSAN effectively encodes raw data into feature representations that reflect degradation patterns; (2) the generated HIs exhibit superior monotonicity, correlation, and robustness compared to existing methods; and (3) the proposed HI-weighted subdomain adaptation mechanism achieves high RUL prediction accuracy, with an average MAE of 0.0989 and RMSE of 0.1189 on the XJTU-SY and PRONOSTIA datasets, outperforming state-of-the-art models.

In future work, we plan to construct a dedicated experimental platform to collect bearing vibration data under various operating conditions, spanning the entire lifecycle from healthy to faulty states. This will help assess the generalizability of the proposed method beyond publicly available datasets. Moreover, to facilitate real-world deployment, we aim to improve the model’s inference speed, ensure compatibility with edge devices, and explore feedback-based retraining strategies to support continuous learning in dynamic industrial environments.

## Figures and Tables

**Figure 1 sensors-25-04536-f001:**
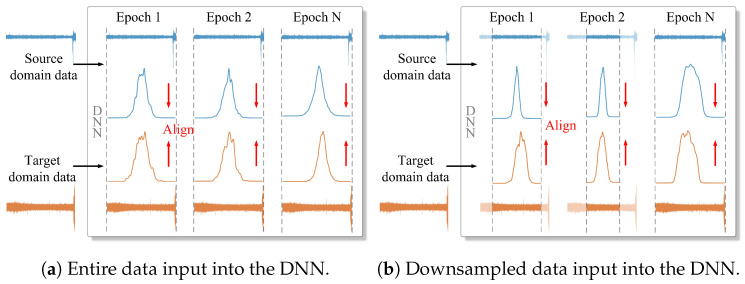
Comparison of data input forms. (**a**) The blue curve and orange curve represent the probability density functions of source domain and target domain data, respectively. The red arrows represent minimizing the discrepancies between source and target domains. (**b**) The data within dashed segments represent source and target domain data downsampled at the current epoch.

**Figure 2 sensors-25-04536-f002:**
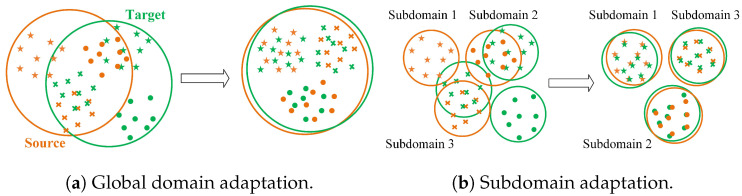
Minimizing domain discrepancies.

**Figure 3 sensors-25-04536-f003:**
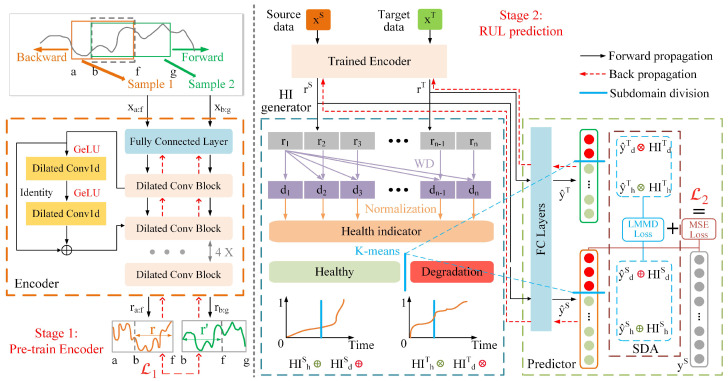
The architecture of the proposed HIWSAN.

**Figure 4 sensors-25-04536-f004:**
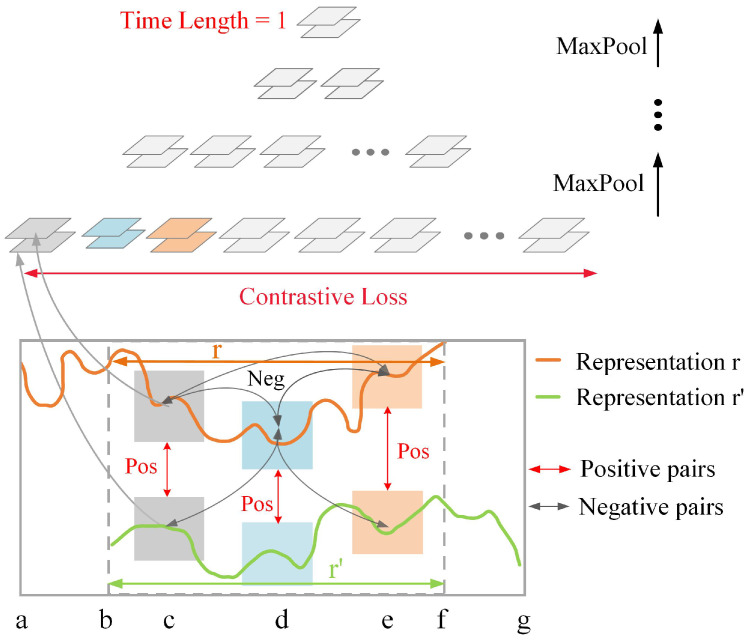
The process of constructing positive–negative pairs and calculating the contrastive loss, where a,b,…,g denote different time steps.

**Figure 5 sensors-25-04536-f005:**
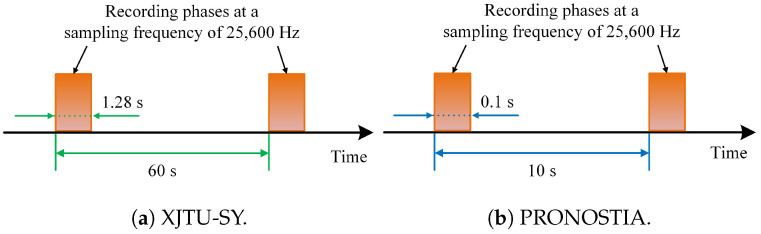
Sampling schemes for two bearing datasets.

**Figure 6 sensors-25-04536-f006:**
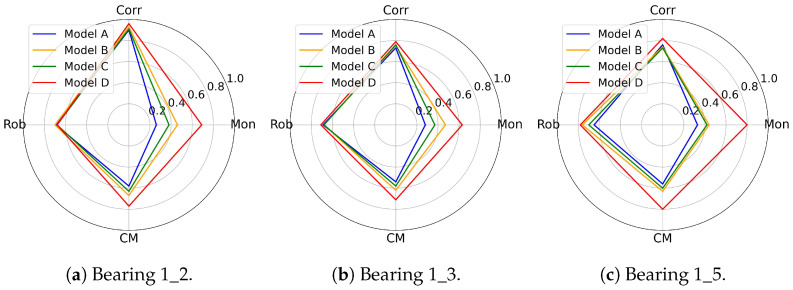
The ablation experiment results for the four models in Table 6.

**Figure 7 sensors-25-04536-f007:**
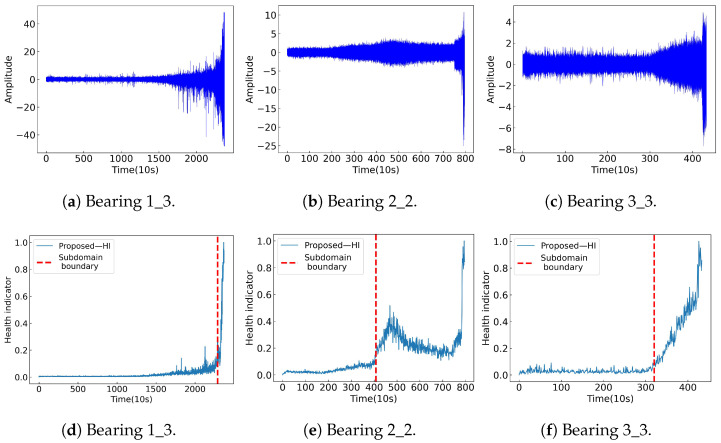
The time-domain curves, HIs, and their subdomains of three test bearings.

**Figure 8 sensors-25-04536-f008:**
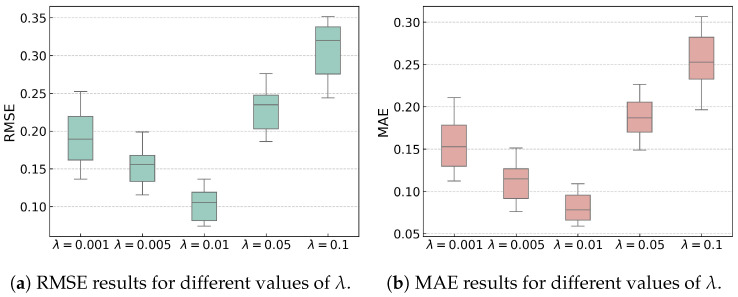
RMSE and MAE results for different tradeoff parameters λ.

**Figure 9 sensors-25-04536-f009:**
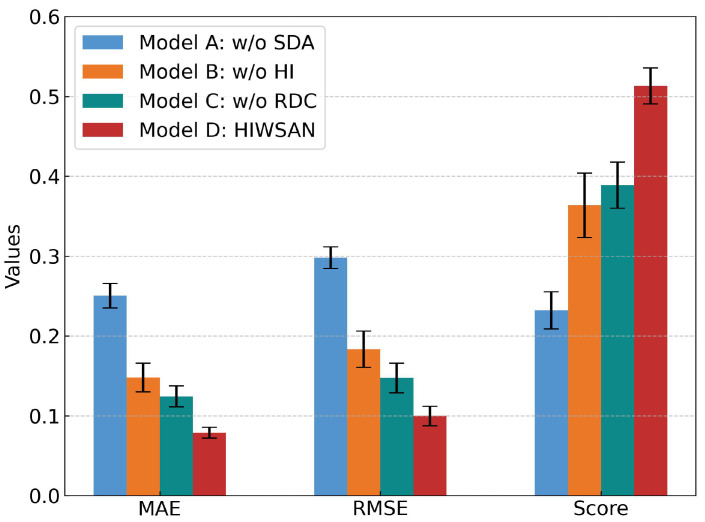
Comparing the four models in Table 11 by MAE, RMSE and Score (defined in Equation (14)).

**Figure 10 sensors-25-04536-f010:**
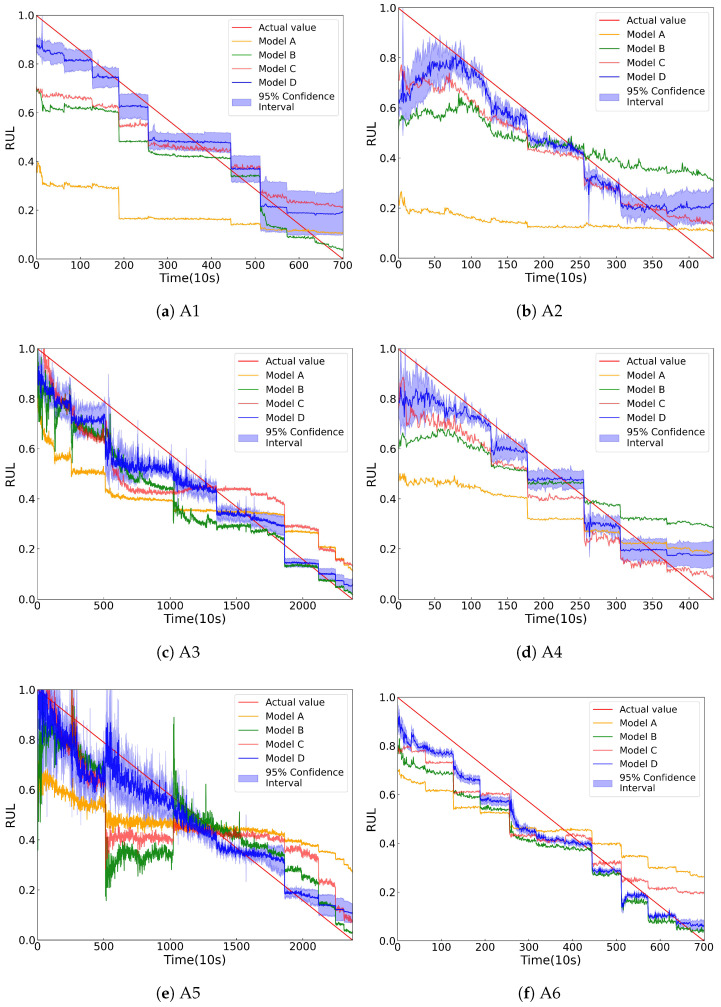
Prediction results for the six transfer tasks in Table 10.

**Figure 11 sensors-25-04536-f011:**
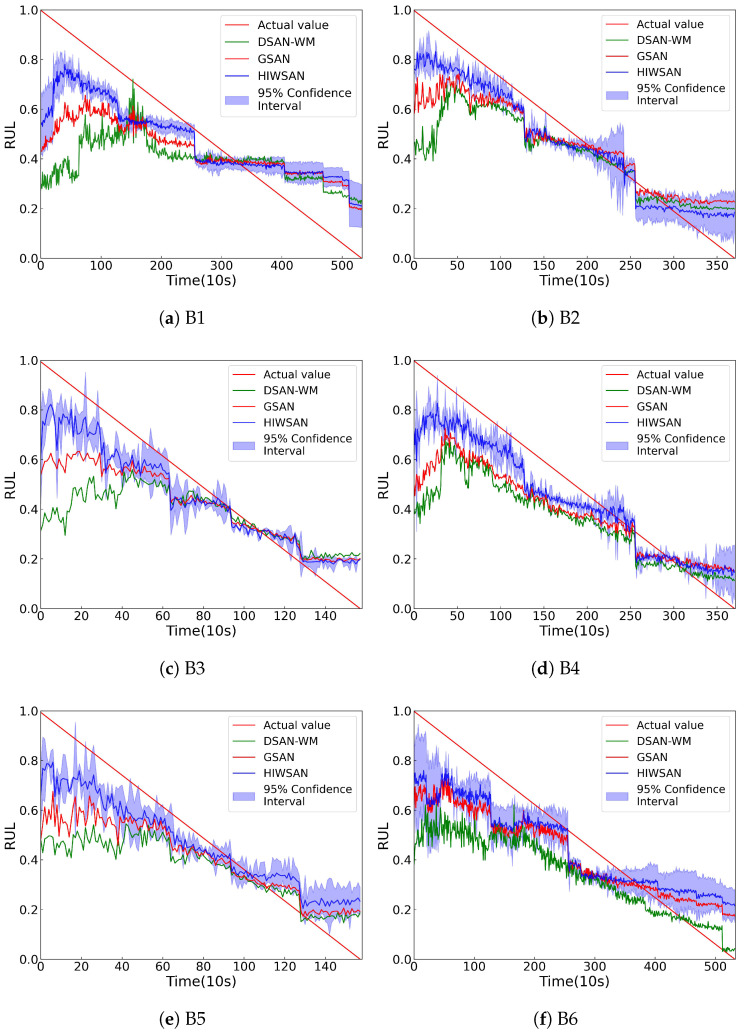
Prediction results for the six transfer tasks in Table 13.

**Figure 12 sensors-25-04536-f012:**
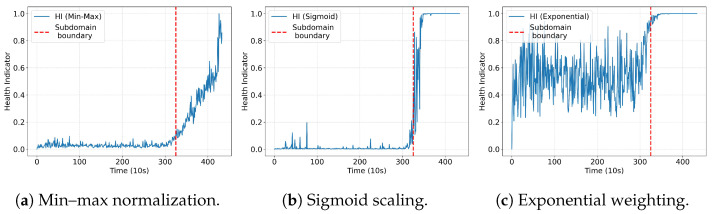
Comparison of different normalization methods.

**Figure 13 sensors-25-04536-f013:**
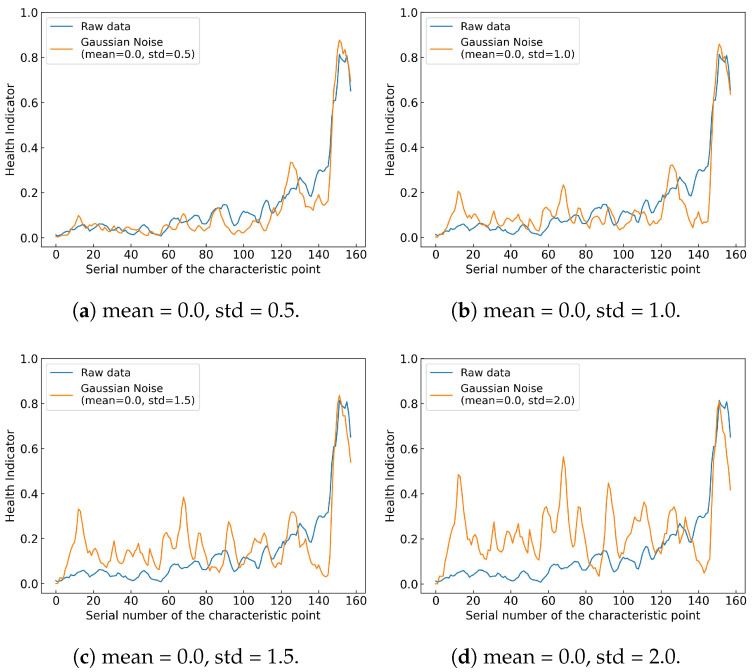
Impact of additive Gaussian noise, with standard deviation ranging from 0.5 to 2.0, on the HI curves.

**Table 1 sensors-25-04536-t001:** List of frequently used symbols and their definitions.

Symbol	Definition
$f_{\theta }$	Nonlinear embedding function *f* associated with network parameters θ
*T*	Number of time steps in the full-life of a bearing
*F*	Number of features in raw data
*M*	Number of features in a feature representation (M<F)
*x*	Full-life time series data of any bearing
$x_{a}$	*a*th data point in the time series data (1≤a≤T)
$x_{a:g}$	*a*-to-*g* segment of the time series data (1≤a,g≤T)
$r_{a:g}$	Feature representations generated by the Encoder
${\tilde{y}}^{S},y^{S}$	Source domain RUL prediction values and ground truth values, respectively
$N^{S},N^{T}$	Number of data points in source and target domains, respectively
${\mathrm{HI}}^{S},{\mathrm{HI}}^{T}$	Health indicators for source and target domain data

**Table 2 sensors-25-04536-t002:** The XJTU-SY datasets [41].

	Condition 1	Condition 2	Condition 3
Load (N)	12,000	11,000	10,000
Speed (rpm)	2100	2250	2400
Bearings	B1_1∼B1_5	B2_1∼B2_5	B3_1∼B3_5

**Table 3 sensors-25-04536-t003:** The PRONOSTIA datasets [42].

	Condition 1	Condition 2	Condition 3
Load (kN)	4	4.2	5
Speed (rpm)	1800	1650	1500
Bearings	B1_1∼B1_7	B2_1∼B2_7	B3_1∼B3_3

**Table 4 sensors-25-04536-t004:** Hyperparameter settings for both pre-training and main training stages.

Hyperparameters	Settings
Optimizer	Adam
Initial learning rate	0.001
Scheduler	StepLR (50, 0.1)
Batch size	1
Max epochs	200
Number of subdomains	2
tradeoff parameter λ	0.01

**Table 5 sensors-25-04536-t005:** The architectural parameters of the proposed Encoder. Note a kernel size of 3 is common among deep learning architectures for a good tradeoff between modeling capacity and efficiency; for example, see [37].

Type	Shape	In/Out Features	Kernel	Dilation	Padding
Linear	(1, 161, 32,768) → (1, 161, 64)	32,768/64	−	−	−
Transpose	(1, 161, 64) → (1, 64, 161)	−	−	−	−
Dilated Conv	(1, 64, 161) → (1, 64, 161)	Block 1: 64/64	3	1	1
Dilated Conv	(1, 64, 161) → (1, 64, 161)	Block 2: 64/64	3	2	2
Dilated Conv	(1, 64, 161) → (1, 64, 161)	Block 3: 64/64	3	4	4
Dilated Conv	(1, 64, 161) → (1, 64, 161)	Block 4: 64/64	3	8	8
Dilated Conv	(1, 64, 161) → (1, 64, 161)	Block 5: 64/64	3	16	16
Dilated Conv	(1, 64, 161) → (1, 64, 161)	Block 6: 64/64	3	32	32
Dilated Conv	(1, 64, 161) → (1, 320, 161)	Block 7: 64/320	3	64	64
Transpose	(1, 320, 161) → (1, 161, 320)	−	−	−	−

**Table 6 sensors-25-04536-t006:** The performance of HIs is constructed from four models.

Bearings	Metrics	Model A	Model B	Model C	Model D
		w/o RS	w/o RDC	w/o HC	-
Bearing 1_2	Mon	0.2634 ± 0.0412	0.4625 ± 0.0587	0.3745 ± 0.0613	**0.6892 ± 0.0411**
	Corr	0.9023 ± 0.0321	0.9275 ± 0.0286	0.9112 ± 0.0341	**0.9598 ± 0.0223**
	Rob	0.6894 ± 0.0542	**0.7024 ± 0.0426**	0.6822 ± 0.0412	0.6777 ± 0.0489
	CM	0.5778 ± 0.0214	0.6756 ± 0.0374	0.6344 ± 0.0294	**0.7732 ± 0.0301**
Bearing 1_3	Mon	0.2845 ± 0.0621	0.4756 ± 0.0653	0.3798 ± 0.0552	**0.6332 ± 0.0214**
	Corr	0.7311 ± 0.0257	0.7542 ± 0.0321	0.7693 ± 0.0546	**0.7974 ± 0.0385**
	Rob	0.6859 ± 0.0275	0.6771 ± 0.0392	0.6794 ± 0.0401	**0.7097 ± 0.0279**
	CM	0.5452 ± 0.0385	0.6173 ± 0.0432	0.5812 ± 0.0418	**0.7094 ± 0.0275**
Bearing 1_5	Mon	0.3297 ± 0.0348	0.4396 ± 0.0187	0.4181 ± 0.0212	**0.8015 ± 0.0584**
	Corr	0.7632 ± 0.0397	0.7439 ± 0.0512	0.7326 ± 0.0601	**0.8231 ± 0.0421**
	Rob	0.6503 ± 0.0559	0.7651 ± 0.0223	0.6998 ± 0.0514	**0.7841 ± 0.0239**
	CM	0.5582 ± 0.0415	0.6283 ± 0.0197	0.6043 ± 0.0184	**0.8006 ± 0.0276**

**Table 7 sensors-25-04536-t007:** Experimental results for comparing proposed-HI with related HI construction methods.

Methods	Metrics	Bearing 2_2	Bearing 2_3	Bearing 2_5
PCA-HI	Mon	0.3412 ± 0.0000	0.4098 ± 0.0000	0.5904 ± 0.0000
	Corr	0.9103 ± 0.0000	0.7397 ± 0.0000	0.8396 ± 0.0000
	Rob	0.6301 ± 0.0000	0.5095 ± 0.0000	0.5710 ± 0.0000
	CM	0.5899 ± 0.0000	0.5411 ± 0.0000	0.6602 ± 0.0000
ISOMAP-HI	Mon	0.2897 ± 0.0000	0.4512 ± 0.0000	0.5294 ± 0.0000
	Corr	0.8698 ± 0.0000	0.6995 ± 0.0000	0.8704 ± 0.0000
	Rob	0.7512 ± 0.0000	0.7403 ± 0.0000	**0.7402 ± 0.0000**
	CM	0.6105 ± 0.0000	0.6109 ± 0.0000	0.6898 ± 0.0000
AE-HI	Mon	0.3996 ± 0.0421	0.4705 ± 0.0399	0.6291 ± 0.0592
	Corr	0.9184 ± 0.0215	0.8798 ± 0.0183	0.8692 ± 0.0227
	Rob	0.7489 ± 0.0174	0.6105 ± 0.0612	0.6613 ± 0.0385
	CM	0.6610 ± 0.0513	0.6397 ± 0.0305	0.7108 ± 0.0497
MCAN-HI	Mon	0.5108 ± 0.0516	0.4612 ± 0.0433	0.6322 ± 0.0387
	Corr	0.9215 ± 0.0314	0.8897 ± 0.0409	0.8799 ± 0.0298
	Rob	0.7405 ± 0.0433	0.7191 ± 0.0197	0.6697 ± 0.0426
	CM	0.6997 ± 0.0208	0.6593 ± 0.0199	0.7201 ± 0.0210
MSMHA-HI	Mon	0.5198 ± 0.0527	0.4793 ± 0.0512	0.6211 ± 0.0484
	Corr	0.9002 ± 0.0211	0.8699 ± 0.0302	0.8897 ± 0.0220
	Rob	**0.7903 ± 0.0398**	0.7298 ± 0.0417	0.7293 ± 0.0341
	CM	0.7213 ± 0.0191	0.6697 ± 0.0210	0.7308 ± 0.0205
Proposed-HI	Mon	**0.6299 ± 0.0621**	**0.5305 ± 0.0412**	**0.6914 ± 0.0428**
	Corr	**0.9195 ± 0.0297**	**0.8897 ± 0.0205**	**0.9192 ± 0.0522**
	Rob	0.7007 ± 0.0203	**0.7408 ± 0.0384**	0.7310 ± 0.0420
	CM	**0.7401 ± 0.0213**	**0.7102 ± 0.0348**	**0.7698 ± 0.0259**

**Table 8 sensors-25-04536-t008:** Generalization performance results averaged over three bearings.

Method	Mon	Corr	Rob	CM
MCAN-HI	0.3016	0.7322	0.7436	0.5634
MSMHA-HI	0.3920	0.7484	0.6151	0.5746
Proposed-HI	**0.4585**	**0.7833**	**0.7564**	**0.6348**

**Table 9 sensors-25-04536-t009:** The architectural parameters of the proposed RUL prediction model.

Type	Shape	Parameters
Encoder	(1, 2375, 2560) → (1, 2375, 320)	in_features = 2560; out_features = 320
Squeeze	(1, 2375, 320) → (2375, 320)	−
Linear	(2375, 320) → (2375, 128)	in_features = 320; out_features = 128
ReLU	(2375, 128)	−
Linear	(2375, 128) → (2375, 1)	in_features = 128; out_features = 1

**Table 10 sensors-25-04536-t010:** The six transfer tasks of the PRONOSTIA dataset.

Task	Conditions	Training Bearings	Test Bearings
A1	C1→C2	Labeled: B1_1; Unlabeled: B2_1	B2_2
A2	C1→C3	Labeled: B1_1; Unlabeled: B3_1	B3_3
A3	C2→C1	Labeled: B2_1; Unlabeled: B1_1	B1_3
A4	C2→C3	Labeled: B2_1; Unlabeled: B3_1	B3_3
A5	C3→C1	Labeled: B3_1; Unlabeled: B1_1	B1_3
A6	C3→C2	Labeled: B3_1; Unlabeled: B2_1	B2_2

**Table 11 sensors-25-04536-t011:** The RUL prediction metrics of Models A-D for the six transfer tasks in Table 10.

Tasks	Metrics	Model A	Model B	Model C	Model D
		w/o SDA	w/o HI	w/o RDC	Proposed Model
A1	MAE	0.3267 ± 0.0367	0.1386 ± 0.0654	0.1266 ± 0.0256	**0.0797 ± 0.0231**
	RMSE	0.3858 ± 0.0406	0.1657 ± 0.0759	0.1622 ± 0.0364	**0.0957 ± 0.0247**
	Score	0.1412 ± 0.0306	0.3778 ± 0.1013	0.4025 ± 0.0428	**0.4640 ± 0.0775**
A2	MAE	0.3725 ± 0.0285	0.1860 ± 0.0439	0.1076 ± 0.0314	**0.0908 ± 0.0131**
	RMSE	0.4450 ± 0.0310	0.2268 ± 0.0496	0.1289 ± 0.0436	**0.1239 ± 0.0205**
	Score	0.1155 ± 0.0216	0.3077 ± 0.0710	0.4618 ± 0.0624	**0.5256 ± 0.0346**
A3	MAE	0.2009 ± 0.0747	0.1267 ± 0.0313	0.1353 ± 0.0159	**0.0824 ± 0.0196**
	RMSE	0.2369 ± 0.0942	0.1624 ± 0.0276	0.1533 ± 0.0201	**0.1036 ± 0.0248**
	Score	0.2830 ± 0.0940	0.4052 ± 0.0691	0.3082± 0.0888	**0.5053 ± 0.0704**
A4	MAE	0.2114 ± 0.0158	0.1577 ± 0.0702	0.1076 ± 0.0100	**0.0708 ± 0.0111**
	RMSE	0.2549 ± 0.0197	0.1867 ± 0.0785	0.1230 ± 0.0116	**0.0891 ± 0.0116**
	Score	0.2617 ± 0.0261	0.3284 ± 0.1246	0.4417 ± 0.0299	**0.5414 ± 0.0212**
A5	MAE	0.2168 ± 0.0520	0.1574 ± 0.0575	0.1555 ± 0.0123	**0.0766 ± 0.0189**
	RMSE	0.2608 ± 0.0580	0.2241 ± 0.0635	0.1849 ± 0.0142	**0.1006 ± 0.0221**
	Score	0.2640 ± 0.0534	0.3274 ± 0.1306	0.2838 ± 0.0123	**0.4839 ± 0.0488**
A6	MAE	0.1740 ± 0.0740	0.1151 ± 0.0422	0.1140 ± 0.0173	**0.0726 ± 0.0118**
	RMSE	0.2046 ± 0.0849	0.1345 ± 0.0191	0.1323 ± 0.0469	**0.0856 ± 0.0143**
	Score	0.3265 ± 0.1220	0.4346 ± 0.1073	0.4348 ± 0.0513	**0.5582 ± 0.0285**

**Table 12 sensors-25-04536-t012:** Experimental results of comparison with related RUL prediction methods.

Type	Model	MAE	RMSE	Score
Baseline	TCNN [8]	0.1961	0.2344	0.2876
	WD-WDANN [47]	0.1871	0.2161	0.3031
DA	MADA [48]	0.1135	0.1409	0.3752
	MCDA [49]	0.1086	0.1367	0.4421
SDA	DSAN-WM [34]	0.1049	0.1352	0.4464
	GSAN [50]	0.0937	0.1154	0.4739
Proposed	HIWSAN	**0.0788**	**0.0998**	**0.5131**

**Table 13 sensors-25-04536-t013:** The six transfer tasks of the XJTU-SY dataset.

Task	Conditions	Training Bearings	Test Bearings
B1	C1→C2	Labeled: B1_1; Unlabeled: B2_1	B2_3
B2	C1→C3	Labeled: B1_1; Unlabeled: B3_3	B3_1
B3	C2→C1	Labeled: B2_1; Unlabeled: B1_3	B1_1
B4	C2→C3	Labeled: B2_1; Unlabeled: B3_3	B3_1
B5	C3→C1	Labeled: B3_1; Unlabeled: B1_3	B1_1
B6	C3→C2	Labeled: B3_1; Unlabeled: B2_3	B2_1

**Table 14 sensors-25-04536-t014:** The RUL prediction metrics of three SDA models for the six transfer tasks in Table 13.

Tasks	Metrics	DSAN-WM	GSAN	HIWSAN
B1	MAE	0.1763 ± 0.0109	0.1595 ± 0.0251	**0.1382 ± 0.0161**
	RMSE	0.2211 ± 0.0236	0.2012 ± 0.0359	**0.1725 ± 0.0255**
	Score	0.3290 ± 0.0294	0.3498 ± 0.0325	**0.3887 ± 0.0364**
B2	MAE	0.1517 ± 0.0178	0.1289 ± 0.0249	**0.0863 ± 0.0117**
	RMSE	0.2084 ± 0.0459	0.1623 ± 0.0460	**0.1017 ± 0.0332**
	Score	0.3735 ± 0.0257	0.3831 ± 0.0495	**0.4821 ± 0.0310**
B3	MAE	0.2102 ± 0.0383	0.1704 ± 0.0563	**0.1090 ± 0.0143**
	RMSE	0.2930 ± 0.0642	0.2297 ± 0.0419	**0.1327 ± 0.0225**
	Score	0.3317 ± 0.0306	0.3442 ± 0.0605	**0.4440 ± 0.0331**
B4	MAE	0.1804 ± 0.0493	0.1608 ± 0.0151	**0.1377 ± 0.0213**
	RMSE	0.2291 ± 0.0645	0.2096 ± 0.0285	**0.1703 ± 0.0249**
	Score	0.3308 ± 0.0801	0.3517 ± 0.0327	**0.3863 ± 0.0451**
B5	MAE	0.1927 ± 0.0521	0.1623 ± 0.0407	**0.1114 ± 0.0155**
	RMSE	0.2609 ± 0.0564	0.2107 ± 0.0373	**0.1320 ± 0.0264**
	Score	0.3501 ± 0.0487	0.3749 ± 0.0602	**0.4295 ± 0.0362**
B6	MAE	0.1812 ± 0.0190	0.1597 ± 0.0296	**0.1311 ± 0.0230**
	RMSE	0.2320 ± 0.0804	0.2002 ± 0.0443	**0.1586 ± 0.0313**
	Score	0.3252 ± 0.0303	0.3550 ± 0.0390	**0.3927 ± 0.0478**

**Table 15 sensors-25-04536-t015:** Impact of additive Gaussian noise, with standard deviation ranging from 0.5 to 2.0, on the HI metrics.

Metric	Raw Data	Mean = 0.0,	Mean = 0.0,	Mean = 0.0,	Mean = 0.0,
		std = 0.5	std = 1.0	std = 1.5	std = 2.0
Mon	0.6688	0.6943	0.3503	0.3121	0.1975
Corr	0.7512	0.6727	0.5727	0.5164	0.4981
Rob	0.7957	0.6982	0.7183	0.7091	0.7246
CM	0.7316	0.6890	0.5274	0.4925	0.4458

## Data Availability

The data presented in this study are openly available through GitHub at https://github.com/WangBiaoXJTU/xjtu-sy-bearing-datasets (accessed on 20 July 2025), reference number [41]; and through the NASA Prognostics Data Repository at https://phm-datasets.s3.amazonaws.com/NASA/10.+FEMTO+Bearing.zip (accessed on 20 July 2025), reference number [42].

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
