# Peer review of "Remaining Useful Life Prediction Across Conditions Based on a Health Indicator-Weighted Subdomain Alignment Network"

_sensors, 2025, doi:10.3390/s25154536_

Round 1
Reviewer 1 Report
Comments and Suggestions for Authors
This manuscript proposes a novel method for RUL prediction across operating conditions based on a health indicator-weighted subdomain alignment network. The proposed method demonstrates high accuracy and good performance. The article features a complete structure and substantial workload. It is recommended for acceptance after addressing the following two issues:
-
The setup of key parameters for the RUL prediction model (such as time window, time interval, and prediction horizon) should be clearly stated, as these parameters form the foundation of the model.
-
The descriptions of the model architecture appear inconsistent throughout the manuscript. For example:
-
- Section 1 mentions "an Encoder, an HI generator, and a subdomain adaptation (SDA) module";
-
The last section describes "an Encoder, an HI generator, a Predictor, and a SDA module";
-
Figure 3 only shows "an Encoder, an HI generator, and a Predictor"
These discrepancies should be reconciled for clarity and consistency.
Reviewer 2 Report
Comments and Suggestions for Authors
This paper proposes a RUL prediction model for bearing. The paper is well-written except the following concerns.
1) In experiment section, it is suggested to list the parameters used for the proposed model in a table which makes the proposed model repeatable.
2) In Abstract, it is suggested to highlight the improvements of the proposed model performance compared with the benchmarks.
3) It is suggested to indicate the full-names when the abbreviations are used for the first time in the main body of the manuscript, e.g. 'RUL'.
Reviewer 3 Report
Comments and Suggestions for Authors
This paper presents a network model for RUL prediction of rolling bearing. Although this paper is generally well written, the topic has been investigated very thoroughly, the contribution and novelty need to be further enhanced. The key issues are as follows:
As I said, this research field has been studied a lot. The introduction is not solid, which doesn’t include the latest and typical works, such as [Transformer-based novel framework for remaining useful life prediction of lubricant in operational rolling bearings], [A wavelet neural network informed by time-domain signal preprocessing for bearing remaining useful life prediction], and [An intelligent hybrid deep learning model for rolling bearing remaining useful life prediction]. It is suggested to benchmark these studies and make the innovation stand out.
The datasets are both public. A bit concerned about the experimental contribution even there were papers published using all public datasets.
Not sure why the titles of two case studies are different, as the final goal as the paper title indicates is to conduct the RUL prediction. It is very unclear.
How to choose these HI construction methods for benchmarking?
The results in Figure 7 are hard to interpret. What does the vertical axis mean and its physical meaning? The average value of many items?
In Conclusion, don’t use the symbols circle 1, circle 2…. looks unprofessional
Reviewer 4 Report
Comments and Suggestions for Authors
This paper proposes a novel deep learning model, the Health Indicator-Weighted Subdomain Alignment Network (HIWSAN), designed to improve remaining useful life (RUL) prediction for bearings under varying working conditions. The model integrates multi-scale feature extraction, health indicator (HI) generation, and subdomain alignment to address key limitations in current domain adaptation methods. Two comprehensive case studies, using the XJTU-SY and PRONOSTIA datasets, are conducted to evaluate the performance of HIWSAN through ablation studies, comparisons with existing models, and generalization tests.
Find attached comments suggested for the improvement of the paper

The manuscript is generally understandable, but there are some areas where sentence structure, grammar, and phrasing could be improved to enhance clarity and flow. A careful language review is recommended to improve overall presentation quality.
Round 2
Reviewer 3 Report
Comments and Suggestions for Authors
The authors provide the reasonable responses to my comments. I don't have further comment.